# The Role of Colchicine in Atherosclerosis: From Bench to Bedside

**DOI:** 10.3390/pharmaceutics14071395

**Published:** 2022-07-01

**Authors:** Leticia González, Juan Francisco Bulnes, María Paz Orellana, Paula Muñoz Venturelli, Gonzalo Martínez Rodriguez

**Affiliations:** 1Centro de Imágenes Biomédicas, Departamento de Radiología, Escuela de Medicina, Pontificia Universidad Católica de Chile, Santiago 8331150, Chile; leticia.gonzalez@uc.cl; 2Instituto Milenio de Ingeniería e Inteligencia Artificial para la Salud, iHEALTH, Pontificia Universidad Católica de Chile, Santiago 7820436, Chile; 3División de Enfermedades Cardiovasculares, Pontificia Universidad Católica de Chile, Santiago 8331150, Chile; jfbulnes@gmail.com (J.F.B.); mporella@uc.cl (M.P.O.); 4Centro de Estudios Clínicos, Instituto de Ciencias e Innovación en Medicina (ICIM), Facultad de Medicina Clínica Alemana, Universidad de Desarrollo, Santiago 7610658, Chile; paumunoz@udd.cl; 5The George Institute for Global Health, Faculty of Medicine, University of New South Wales, Sydney, NSW 2042, Australia

**Keywords:** atherosclerosis, inflammation, colchicine, NLRP3 inflammasome, coronary artery disease, acute coronary syndrome

## Abstract

Inflammation is a key feature of atherosclerosis. The inflammatory process is involved in all stages of disease progression, from the early formation of plaque to its instability and disruption, leading to clinical events. This strongly suggests that the use of anti-inflammatory agents might improve both atherosclerosis progression and cardiovascular outcomes. Colchicine, an alkaloid derived from the flower *Colchicum autumnale*, has been used for years in the treatment of inflammatory pathologies, including Gout, Mediterranean Fever, and Pericarditis. Colchicine is known to act over microtubules, inducing depolymerization, and over the NLRP3 inflammasome, which might explain its known anti-inflammatory properties. Recent evidence has shown the therapeutic potential of colchicine in the management of atherosclerosis and its complications, with limited adverse effects. In this review, we summarize the current knowledge regarding colchicine mechanisms of action and pharmacokinetics, as well as the available evidence on the use of colchicine for the treatment of coronary artery disease, covering basic, translational, and clinical studies.

## 1. Introduction

Cardiovascular (CV) diseases remain the leading cause of mortality worldwide, accounting for up to one third of all registered deaths [1]. The main underlying cause behind cardiovascular disease is atherosclerosis, a chronic inflammatory disease targeting large and medium-sized arteries. In the USA alone, 400,000 people die of coronary artery atherosclerosis and over one million suffer from acute coronary syndrome each year [2]. Management of cardiovascular disease focuses on three main areas: (i) lipid lowering strategies, (ii) control of non-lipid risk factors (such as diabetes, hypertension, obesity, etc.) and (iii) stabilization of the atheromatous plaque, preventing rupture and thrombosis [3]. Current therapies, however, fail to prevent the reoccurrence of ischemic events, a phenomenon known as residual risk [4]. In a ten-year follow-up study of patients post ST elevation myocardial infarction (STEMI), 42% of patients presented with recurrent ischemic events, a risk that was highest during the first year (23.5% per patient/year) even when receiving currently recommended pharmacological treatment [5]. Therefore, the need for new therapies has shifted the focus towards anti-inflammatory drugs than can potentially target the chronic inflammation milieu of atherosclerotic plaques [6].

## 2. Methods

We conducted a full search of animal and human research, from basic studies to randomized clinical trials and meta-analyses, examining the use of colchicine for the treatment of atherosclerosis and/or coronary artery disease. Medline, Pubmed and Embase databases were searched until May 2022. Two researchers independently screened titles and abstracts of articles for full-text review. After data extraction, two researchers chose the most relevant articles and were in charge of elaborating the initial text, which was then sent to every author for further evaluation. If there were any discrepancies on a specific subject, the topic was re-analyzed, and a consensus was achieved. The final version of the manuscript was approved by every author.

## 3. Inflammation in Atherosclerosis

Inflammation plays a central role in the pathogenesis of atherosclerosis [7]. Both the innate and adaptative immune responses are involved in the process of atheroma formation, with monocyte/macrophages as key players throughout disease progression [8]. The development of the atherosclerotic plaque starts with the infiltration and accumulation of modified, apolipoprotein B-containing lipoproteins within the vessel wall [9]. Once in the intima layer, oxidized cholesterol in lipoproteins triggers the activation and production of inflammatory mediators in charge of recruiting circulating monocytes to the site of injury [9]. Within the vessel wall, monocytes differentiate to macrophages and engulf modified lipoproteins, becoming foam cells [9]. Foam cells continue to release inflammatory cytokines, in particular TNF-α and interleukin-1β (IL-1β) [10], exacerbating endothelial dysfunction and perpetuating the inflammatory response [11]. Advanced atherosclerotic plaques are characterized by a large lipid-rich core—composed of foam cells, cell debris and extracellular cholesterol—and a fibrous cap, formed by extracellular matrix and smooth muscle cells [12]. In later stages of the disease, macrophages within the plaque release matrix metalloproteinases that target the fibrous cap, destabilizing the plaque and setting the stage for plaque rupture and the consequent ischemic event [13,14] (Figure 1).

Neutrophils also participate in all stages of atherosclerosis development [15]. Indeed, circulating levels of neutrophils in humans predict future cardiovascular events [16], while in mice they correlate with the size of the developing plaque [17]. Myeloperoxidase (MPO), the main component of neutrophil granules, has been found in atherosclerotic plaques [18]. It has been shown that MPO-induced lipid peroxidation favors foam cell formation [19]. MPO can also activate metalloproteinases, inducing plaque disruption, by the release of reactive oxygen species (ROS) [20]. Similarly, neutrophils are known to release extracellular matrix proteinases that contribute to plaque destabilization [21], like elastase [22] and proteinase-3 [23], locating particularly in rupture-prone areas of the plaque [24,25]. Neutrophil depletion in apolipoprotein E knockout (ApoE KO) mice has been shown to reduce monocyte infiltration and plaque formation in the aorta [26]. In fact, neutrophils can affect monocyte recruitment through several mechanisms [27], including increased expression of adhesion molecules in the endothelium through the release of granule-proteins proteinase 3 and azurocidin [28]. Furthermore, neutrophil extracellular traps (NETs) are web-like structures made of genetic material, histones, MPO and others, which are released upon neutrophil activation [29]. The process of NETs formation is called NETosis and is introduced to discriminate this pathway from other types of cell death [30]. Of note, NETs have been found in atherosclerotic plaques of both mice and humans [31,32,33]. Increased levels of NETosis markers are associated with the severity of coronary atherosclerosis in patients [34]. Similarly, Mangold et al. have shown that the number of NETs and activated neutrophils in patients with acute coronary syndrome (ACS) is related to final infarct size [35]. Finally, autopsied disrupted plaques (i.e., with hemorrhage or erosion) from patients with ACS, presented significantly more neutrophils and NETs compared with plaques without those features [36].

Cholesterol accumulates in atherosclerotic plaques not only in the form of intracellular cholesterol esters but also intra and extracellular cholesterol crystals (CCs) [37]. CCs have been found in all stages of atherosclerotic plaque development [38,39] and have been shown to be associated with plaque rupture [40]. CCs can induce NETosis, which can in turn prime macrophages to synthetize cytokine precursors, such as pro-IL-1β [31]. CCs can also directly activate the nucleotide-binding oligomerization domain-like receptor, pyrin domain-containing 3 (NLRP3) inflammasome in macrophages, resulting in the release of mature IL-1β, a key cytokine involved in the development and disruption of atherosclerotic plaques [37].

## 4. The NLRP3 Inflammasome

The innate immune system relies on a set of germline-encoded pattern recognition receptors (PRRs) to recognize pathogenic insults as well as defective cells [41]. PRRs detect the presence of pathogen-associated molecular patterns (PAMPs) or damage-associated molecular patterns (DAMPs), triggering downstream inflammatory pathways leading to removal of the insult and tissue repair [41]. The inflammasomes are cytosolic, multimeric protein complexes that respond to PAMPs and DAMPs [42]. Five members of the PRRs have been confirmed to form inflammasomes, key among them the NLRP3 inflammasome [43]. The NLRP3 inflammasome responds to a wide variety of activators including monosodium urate, extracellular adenosine triphosphate (ATP) and CCs [44]. The basal levels of the components of the NLRP3 inflammasome and their targets are very low [45]. As such, a two-step process of priming and activation is required [46,47]. The priming step is induced by the activation of Toll-like receptors (TLRs) and cytokine receptors, leading to upregulation of the transcription of NLRP3 and pro-IL-1β [48]. Afterwards, further stimuli will promote the inflammasome assembly, resulting in cytokine production [45]. Upon activation, the NLRP3 receptor oligomerizes and interacts with the adaptor protein apoptosis-associated speck-like protein containing a caspase-recruitment domain (ASC), to recruit and activate pro-caspase-1 [49,50]. Active caspase-1 can now cleave pro-IL-1β and pro-IL-18 into their biologically active, highly inflammatory forms [42].

In the context of atherosclerosis, CCs are considered a major driver of NLRP3 inflammasome activation [37]. It has been postulated that CCs activate the inflammasome in a process involving lysosomal damage after phagocytosis [51]. The inefficient clearance of CCs results in the leakage of cathepsin B into the cytoplasm, which in turn can activate the inflammasome complex [52]. Accordingly, the inflammatory response elicited by intraperitoneal CCs injection in WT mice is absent in mice deficient in NLRP3 inflammasome components [38]. Oxidized LDL has also been shown to induce NLRP3 activation through lysosomal disruption via interaction with CD36 [53]. As such, macrophages lacking CD36 fail to release IL-1β, indicating lack of NLRP3 activation [53].

The contribution of IL-1β to atherogenesis has been established by several animal studies, hinting at a role of the NLRP3 inflammasome in disease development [54,55,56]. Bone-marrow deficiency of ASC is associated with reduced vascular inflammation and neointimal formation in a mouse model of vascular injury [57]. LDLR KO mice receiving NLRP3, ASC or IL-1β deficient bone marrow developed significantly reduced atherosclerosis when challenged with an atherogenic diet [38]. Similarly, systemic or bone-marrow deficiency of caspase-1/11 is associated with a reduction in atherosclerotic lesions in both ApoE [58,59] and LDLR KO mice [60]. Conversely, bone marrow transplantation studies in ApoE KO mice have shown that lack of NLRP3, ASC or caspase-1 had no effect on atherosclerotic plaque size, plaque stability or macrophage infiltration [61]. Differences in experimental conditions might explain these contrasting results. In humans, high expression of NLRP3 inflammasome components has been detected in carotid atherosclerotic plaques [62] and high levels of expression of NLRP3 correlate with the severity of coronary artery atherosclerosis [63]. The inflammasome has been shown to be primed in peripheral monocytes from ACS patients compared with controls [64] and levels of NLRP3, IL-1β, IL-18 and other inflammasome components have been found to be elevated in ACS patients compared with controls [65]. Taken together, these studies highlight the important role of IL-1β and the NLRP3 inflammasome in atherosclerosis and mark them as possible targets for the development of new therapeutic strategies.

## 5. Colchicine

Colchicine is a botanical alkaloid derived from the flower *Colchicum autumnale*, first described as a medicinal plant in the Ebers papyrus of ancient Egypt in 1550 BC, where it was used for the management of pain and swelling [66]. The colchicine molecule, chemical name N-[(7S)-5,6,7,9-tetrahydro-1,2,3,10-tetramethoxy-9-oxobenzo(a)heptalen-7-yl)acetamide], is composed of three rings [67]. The A (trimethoxyphenyl moiety) and C ring (methoxytropone moiety) are highly involved in binding to tubulin and are maintained in a rigid configuration by B-rings [67]. Modifications to both the A and C rings significantly affect tubulin binding [67,68], while modifications on the B rings are associated with changes in activation energy of the binding and association/dissociation kinetics [69].

Nowadays, colchicine is widely used for the treatment of acute gout flares and Familial Mediterranean Fever (FMF) [70,71]. It has also been used in other inflammatory conditions such as calcium pyrophosphate disease, Adamantiades–Behcet’s syndrome and—in the cardiovascular field—pericarditis [72]. Given the ease of access, low cost and favorable safety profile, colchicine has emerged as a potential oral treatment targeting the inflammatory component of atherosclerosis.

Mechanistically, colchicine acts by inhibiting tubulin polymerization, disrupting the cellular cytoskeleton, and impairing several processes including mitosis, intracellular transport, and phagocytosis [73]. Furthermore, colchicine inhibits neutrophil chemotaxis and adhesion to the inflamed endothelium [74]. At nanoconcentrations, colchicine alters E-selectin distribution on endothelial cells, affecting neutrophil adhesion [75]. On the µM level, colchicine induces L-selectin shedding, preventing recruitment [75]. Paschke and colleagues have also shown that colchicine affects human neutrophil deformability and motility, affecting a key step in inflammatory processes, cell extravasation [76]. Monosodium urate (MSU)-induced superoxide production in neutrophils in vitro is also affected by colchicine treatment [77]. Superoxide production inhibition has also been reported in vivo in MSU-treated peritoneal macrophages, at a dosage significantly lower than the required to affect neutrophil infiltration [78]. Colchicine also modulates TNF-α synthesis by rat liver macrophages [79] and downregulates TNF receptors in both macrophages and endothelial cells [80].

Platelets play a key role in atherosclerosis complications [12]. Early reports have suggested that colchicine might also affect platelet aggregation [81] through mechanisms involving inhibition of cofilin and LIM domain kinase 1 [82]. In FMF patients, colchicine reduced β-thromboglobulin levels, a protein released during platelet activation, with no effect on mean platelet volume, a marker of platelet activity [83]. A more recent experiment by Shah and colleagues showed that colchicine administration to healthy subjects at a clinically relevant dose (1.8 mg single dose) reduced leukocyte-platelet aggregation (both monocyte and neutrophil) as well as levels of surface markers of platelet activity, such as p-selectin and PAC-1 (activated GP IIb/IIIa), with no effect on homotypic platelet aggregation [84]. Likewise, Raju and colleagues have also showed that oral colchicine at a dose of 1 mg did not affect platelet aggregation in response to several stimuli, in patients with acute coronary syndrome [85]. It is possible then that the anti-inflammatory action of colchicine on neutrophils could impact platelet function. NETs can facilitate thrombosis by promoting platelet adhesion, activation, and aggregation, and also the accumulation of prothrombotic factors such as von Willebrand factor and fibrinogen [86]. Interestingly, colchicine has been shown to reduce NETosis induced by CC [87], in neutrophils isolated from individuals with Behcet’s disease [88] and in patients with ACS [89].

## 6. Colchicine and the NLRP3 Inflammasome

Several studies have confirmed that colchicine limits NLRP3 inflammasome activity. Martinon and colleagues first described inactivation of the NLRP3 inflammasome by colchicine in THP1 cells treated with monosodium urate crystals [90]. In patients with FMF colchicine suppressed IL-1β release from both bone marrow-derived macrophages where the inflammasome was constitutively activated and peripheral blood monocytes [91]. Misawa and colleagues also reported a dose-dependent effect of colchicine on IL-1β production in J774 macrophages treated with various inducers of the NLRP3 inflammasome, including MSU and ATP [92]. Furthermore, colchicine has been described as inhibiting the intracellular transport of ASC, preventing co-localization of NLRP3 components and thus the consequent release of active IL-1β [92]. The pathway through which colchicine inhibits the NLRP3 inflammasome is not clear, however some mechanisms have been proposed. In a model of small intestinal injury, Otani and colleagues showed that colchicine inhibited protein expression of cleaved caspase-1 and IL-1β, without affecting mRNA levels of NLRP3 or IL-1β. Monocyte caspase-1 inhibition has been detected in ACS patients as well [64]. Very recently, using molecular dynamic simulation in a mouse model, it has been proposed that colchicine could interact with the ATP binding region of the NLRP3-NACHT domain, thus potentially precluding inflammasome activation [93]. Similarly, in ATP-mediated NLRP3 inflammasome activation, formation of P2X7 pores is a key step [94]. In this context, Marques-da-Silva et al. demonstrated that colchicine is a strong inhibitor of pore formation in mouse peritoneal macrophages after ATP-induced ethidium bromide permeability, resulting in the release of lower levels of ROS and IL-1β [95]. The same results were also seen in vivo, in mice inoculated with lipopolysaccharide and ATP [95].

The ability of colchicine to interfere with the activation of the NLRP3 inflammasome, in addition to its effect on microtubule formation and neutrophil-mediated inflammation, points towards the use of colchicine as a potential strategy to target the inflammatory component of atherosclerosis.

## 7. Pharmacokinetics and Safety

Colchicine is rapidly absorbed by the jejunum and ileum with a bioavailability that fluctuates between 24 to 88% [96]. Peak plasma concentrations occur within 0.5–2 h after oral administration, although it can be detected in leukocytes up to 10 days after ingestion [97]. In circulation, approximately 40% is conjugated to plasmatic proteins and the establishment of colchicine–protein complexes in tissues contributes to its large volume distribution [67]. Colchicine is rapidly distributed to peripheral leukocytes and concentration in these cells may exceed those detected in plasma [98]. Most of the drug undergoes enterohepatic recirculation, leading to a second peak in plasma within 6 h of ingestion, and is eliminated through feces and bile [97]. Around 20% of colchicine is eliminated in the urine at 2 h and 30% after 24 h [97]. Colchicine average elimination half-life is 20 h, which can be prolonged by certain pathologies including renal failure and hepatic cirrhosis [99].

Gastrointestinal (GI) intolerance (diarrhea, abdominal pain, vomiting) is the most common side effect of colchicine, affecting around 20% of patients, followed by myalgias [100]. Most of these side effects can be managed with lower daily doses (around 0.5 mg/day) or long-term treatments [100]. Regarding safety, the first meta-analysis that looked at the safety of colchicine use only reported an 83% increased risk of GI side effects, with no evidence of significant serious adverse effects over 824 patient-years [101]. Similar results have been recently reported by Stewart and colleagues in a meta-analysis of 35 randomized controlled trials [102]. Colchicine significantly increased diarrhea (RR 2.4, 95% confidence interval (CI) 1.6–3.7) and other GI events (RR 1.7, 95% CI 1.3–2.3). No increased rate of other adverse events was detected, including infection, hepatic, hematological, muscular, and sensorial events [102]. On the other hand, a pooled analysis of randomized clinical trials comparing low dose colchicine versus placebo showed that colchicine was associated with a significantly higher risk of non-CV death (OR 1.55; 95% CI 1.10 to 2.17; *p* = 0.01) compared with the placebo group [103]. However, few studies were included and, as stated by the authors, the events reported were low, indicating that the data should be considered with caution [103]. In line with this, a recent retrospective cohort of 24,410 patients with gout showed that colchicine use resulted in an increased risk of pneumonia (adjusted HR, 1.42; 95% CI 1.32–1.53; *p* < 0.05), related to use duration and accumulated dose [104].

Colchicine interacts with two main proteins that impact its pharmacokinetics and pharmacodynamics: cytochrome P450 3A4 (CYP3A4) and P-glycoprotein. Intestinal and hepatic CYP3A4 metabolize colchicine by demethylation, producing 2- and 3-demethylcolchicine [97]. P-glycoprotein, on the other hand, limits GI availability by extruding colchicine from the GI tract [67]. Adverse reactions have been reported by patients consuming either CYP3A4 inhibitors or P-glycoprotein inhibitors, resulting in altered colchicine metabolism and toxicity [67]. Dose adjustment is recommended in these situations. Some cases of myopathy and/or rhabdomyolysis have been reported for instances when colchicine was used in conjunction with statins, however, they are in general well tolerated when used simultaneously [105]. Death due to colchicine administration has been reported when used in conjunction with clarithromycin, particularly in patients with renal insufficiency [106].

## 8. Colchicine in Atherosclerosis

### 8.1. Pre-Clinical Studies

The first animal studies looking at the potential use of colchicine for atherosclerosis showed rather conflicting results. Colchicine administration of high-lipid diet-fed rabbits resulted in a reduction of circulating lipids, restoration of normal triglyceride levels and a protective effect on plaque development in the aorta [107]. On the other hand, colchicine administration on a swine model of balloon-induced atherosclerosis showed the opposite effect, with a mild worsening of plaque development [108]. Though the models used were different, the inconclusive results might have tampered the interest in the usage of this drug for atherosclerosis management. Early in vitro experiments in smooth muscle cells isolated from human atherosclerotic plaques showed that colchicine affected proliferation and migration, which could potentially impact atherosclerosis development [109]. More recently, Huang and colleagues have shown that colchicine administration to hyperlipidemic rats resulted in a reduction in circulating levels of C-reactive protein and lipoprotein associated phospholipase A2, while elevating nitric oxide production, pointing to an improvement in endothelial function. Interestingly, this effect was further enhanced by the concomitant administration of atorvastatin [110]. While Kaminiotis and colleagues have shown that oral administration of colchicine in high-cholesterol diet-fed rabbits showed no effects on atherosclerosis progression [111], Mylonas and colleagues have shown in the same model that colchicine-based anti-inflammatory therapies significantly diminished de novo atherogenesis, decreased triglyceride levels [112,113] and also tampered Krüppel-like factor 4—a transcription factor involved in atheromatosis—overexpression in thoracic aortas [113].

The effect of colchicine in acute myocardial infarction has also been explored. Intraperitoneal administration of colchicine in a mouse model of ischemia/reperfusion resulted in a significant reduction in infarct size 24 h after injury, when administered before reperfusion had been established [114]. Significant improvement in hemodynamic parameters and cardiac fibrosis were also reported [114]. Colchicine has also been shown to be protective when administered after ischemia/reperfusion injury, reducing macrophage infiltration, cardiac remodeling, and dysfunction in a rat model of left coronary artery (LCA) ligation [115]. Similarly, in a mouse model of MI induced by permanent LCA ligation, short-term colchicine administration post MI significantly improved survival, cardiac function and heart failure [116]. This improvement in cardiac performance was associated with a reduction in monocyte and neutrophil infiltration, mRNA expression of inflammatory cytokines and components of the NLRP3 inflammasome 24 h after injury [116]. In SRBI KO/ApoeR61^h/h^ mice, orally administered colchicine resulted in improved survival after feeding with atherogenic diet, however, neither inflammatory markers nor aortic plaque volume were different between the treated and untreated groups (González L, Martínez G, unpublished data).

Finally, the role of colchicine in plaque stabilization has been recently explored by Cecconi and colleagues [117], where atherosclerosis was induced by a high cholesterol diet and balloon endothelial denudation in rabbits. Colchicine treatment reduced the relative increase in aortic wall volume, measured as normalized wall index, and inflammation, measured as 18F-FDG uptake in PET/CT imaging, which could potentially help stabilize the plaque. This effect, however, was only seen in animals with high levels of circulating cholesterol [117]. Table 1 summarizes the available pre-clinical data on colchicine and atherosclerosis.

### 8.2. Translational Studies

In a pilot study including 64 patients with stable coronary artery disease (CAD), Nidorf and colleagues showed that colchicine in low doses decreased high-sensitivity C-reactive protein (hsCRP) levels, a biomarker of inflammation, with no significant side effects [118]. However, the same protective effect was not seen in a pilot randomized controlled trial including patients with acute coronary syndrome (ACS) or stroke, where colchicine failed to reduce hsCRP levels [85]. The different results could be explained by the cause behind hsCRP elevation, which might not be sensitive to colchicine treatment, the dosage used and the context in which the drug was used—acute vs chronic inflammation. A local approach was then used by our group, in which the effect of colchicine on inflammatory cytokine production was assessed in blood samples collected from the coronary sinus [119]. ACS patients were recruited and randomized to receive either colchicine or placebo on top of standard therapy and levels of IL-1β, IL-18 and IL-6 were quantified. Acute colchicine administration resulted in a significant reduction in transcoronary cytokine gradients, suggesting a local intracardiac effect on the NLRP3 inflammasome [119]. In a follow-up study in a different cohort of ACS patients, a significant reduction in the release of IL-1β was observed with colchicine treatment in stimulated peripheral blood monocytes [64]. This reduction was associated with a suppression of monocyte caspase-1 activity. Colchicine was also able to reduce transcoronary and monocyte production of chemokines in treated ACS patients compared with control, which could also positively impact CV outcomes [120]. Furthermore, colchicine treatment suppressed NET production post percutaneous coronary intervention in ACS patients, a process that has been associated with periprocedural MI [89]. Tumor necrosis factor (TNF)-related apoptosis-inducing ligand (TRAIL) is a cytokine belonging to the TNF family of ligands. Evidence shows that a deficiency in circulating TRAIL is associated with atherosclerotic plaque development, probably by inducing a more dysfunctional type of macrophage, with less migratory capacity, and impaired reverse cholesterol efflux and efferocytosis [121]. Research from our group shows that acute colchicine treatment significantly increases plasma TRAIL levels, purportedly regulating the inhibitory effect of IL-18 upon TRAIL [122].

The anti-inflammatory potential of colchicine in chronic CAD has been recently revaluated. Colchicine—either alone or in combination with methotrexate—did not improve coronary endothelial function in patients with stable CAD, measured through non-invasive MRI [123]. In a proteomics study, serum samples from CAD patients were compared before and 30 days after colchicine treatment. The expression of a total of 37 proteins was reduced, including members of the NLRP3 inflammasome pathway (IL-18, IL-1 receptor antagonist and IL-6), adaptative immune system proteins (C-C motif chemokine 17, CD40 ligand, pro-IL-6) and proteins involved in neutrophil degranulation (myeloperoxidase, myeloblastin and azurocidin among others) [124]. A reduction in median hsCRP has also been reported [124]. In the same cohort of patients, colchicine reportedly affected some biomarkers of inflammation. Colchicine treatment for a year resulted in a reduction of extracellular vesicle (EV) NLRP3 protein but no changes in serum NLRP3 protein levels [125]. Lower levels of hsCRP were also detected but this reduction was not related to EV NLRP3 protein levels [125]. MicroRNAs are known to be involved in multiple pathways driving atherosclerosis development. Barraclough and colleagues recently studied the microRNA signature in ACS patients and how colchicine might affect its expression [126]. Plasma samples collected from the aorta, coronary sinus and right atrium were collected from control, ACS standard therapy and ACS standard therapy plus colchicine patients. A total of 30 miRNAs were significantly elevated in the ACS group compared with controls. In patients with ACS, 12 miRNAs were lower when patients received colchicine and seven of these returned to control levels after colchicine treatment. More importantly, three miRNAs suppressed by colchicine are known to be regulators of inflammatory pathways, indicating that levels of miRNAs could potentially be used to track treatment effectiveness [126].

### 8.3. Phase 2 Clinical Studies

Colchicine has been shown to be protective in the context of ischemia/reperfusion, in accordance with pre-clinical results. The perioperative administration of colchicine to patients undergoing coronary bypass grafting resulted in a reduction in postoperative levels of myocardial injury biomarkers such as high-sensitivity troponin T (hsTrop) and creatine kinase-myocardial brain fraction (CKmb) [127]. Deftereos and colleagues have also reported beneficial effects of colchicine administration to STEMI patients treated with percutaneous coronary intervention [128]. Colchicine significantly reduced CKmb concentrations as well as infarct size when compared with placebo group [128]. However, the anti-inflammatory effect of colchicine could not be demonstrated in another study including STEMI patients, where hsCRP levels remained unaffected by the treatment [129]. Oral administration of high dose colchicine also demonstrated no effect on infarct size (assessed by cardiac magnetic resonance) in STEMI patients, when administered at reperfusion and for five consecutive days [130].

The anti-inflammatory effect of colchicine treatment in ACS patients seems to be associated with positive changes at atherosclerotic plaque level. In a prospective nonrandomized observational study including 80 patients with recent ACS, colchicine administration was associated with a reduction in low attenuation plaque volume (LAPV)—a measure of plaque instability and predictor of future coronary events [131]. A positive correlation between LAPV and reduced hsCRP levels has also been reported [131]. Furthermore, the addition of colchicine on top of standard therapy in ACS patients (0.5 mg daily for six months) positively impacted the occurrence of major adverse cardiovascular events, improving overall survival in a randomized, placebo-control trial [132]. Reduction in inflammatory markers (hsCRP, IL-6) after colchicine treatment has been reported in chronic coronary artery disease as well [133].

The Colchicine–PCI randomized trial evaluated the effect of colchicine administration before (1–2 h, 1.8 mg) percutaneous coronary intervention on post-PCI myocardial injury, in patients with stable angina (SA) and ACS [134]. Shah and colleagues have reported that preprocedural colchicine did not protect against PCI-related myocardial injury (including PCI-related MI and MACE at 30 days), despite a reduction in IL-6 and hsCRP levels (22–24 h post-PCI) [134]. Dissimilar results have been reported by Cole and colleagues in a similar pilot study, evaluating SA and ACS patients [135]. Colchicine administration prior to PCI intervention (1.5 mg, 6–24 h) significantly reduced major and minor periprocedural MI and injury, especially in NSTEMI patients [135]. Colchicine also significantly reduced pre-PCI inflammatory cytokine levels (IL-6, IL-1β, TNF-α, IFN-γ) and white blood cell counts, with no differences in post-PCI values [136]. Absolute Troponin change was also reportedly lower in the colchicine group [136]. The difference in results might be influenced by both the population studied and, very importantly, the time of colchicine administration.

### 8.4. Phase 3 Clinical Studies and Meta-Analyses

In the setting of ACS, the COLCOT trial randomized 4745 patients to receive colchicine (0.5 mg BID) or placebo within 30 days post-MI [137]. Colchicine led to a significant reduction of the primary outcome (a composite of death from cardiovascular causes, resuscitated cardiac arrest, myocardial infarction, stroke, or urgent hospitalization for angina leading to coronary revascularization) by 23% (HR 0.77; 95% CI 0.61–0.96; *p* = 0.02). This was mainly driven by a significant reduction in the incidence of stroke (HR 0.26; 95% CI 0.10–0.70) and urgent hospitalization for angina leading to coronary revascularization (HR 0.50; 95% CI 0.31–0.81) [137]. Interestingly, in a post-hoc analysis of COLCOT, time-to-treatment initiation (i.e., length of time between the index MI and the initiation of colchicine) was inversely correlated with colchicine clinical benefit. Indeed, when administered in-hospital within the first three days after the event, colchicine was associated with a 48% reduction in the risk of ischemic events; which contrasted with a lack of benefit when started later (four to seven days, and seven to thirty days) [138]. The other RCT in the ACS setting, the COPS trial, was an Australian-based study that randomly assigned 795 patients diagnosed with MI or unstable angina to receive colchicine (0.5 mg BID for one month, then 0.5 mg QD for eleven months) vs. placebo [139]. Although the original trial failed to demonstrate a benefit on the one-year primary outcome, an extended 24-month follow-up did show a significant 40% reduction in the composite of all-cause mortality, ACS, ischemia-driven-unplanned-urgent revascularization, and non-cardioembolic ischemic stroke. Of note, just as in COLCOT, the main outcome was driven by a significant reduction in urgent revascularization (HR, 0.19; 95% CI 0.05–0.66; *p* = 0.009) [140].

In the setting of chronic CAD, the LoDoCo trial randomized 532 patients to receive colchicine 0.5 mg or no colchicine, using an open label design [141]. Colchicine led to a reduction of the primary outcome (composite of ACS, out-of-hospital cardiac arrest, or non-cardioembolic ischemic stroke) of 67% (HR 0.33; 95% CI 0.18–0.59; *p* < 0.001), due to a significant reduction in the risk of ACS (HR 0.33; 95% CI 0.18–0.63; *p* < 0.001). This same group published, seven years later, the LoDoCo 2 trial, using a more robust—double blinded, placebo controlled—study design and a 10-fold higher number of patients [142]. In this landmark trial, colchicine led to a reduction in the primary outcome (a composite of cardiovascular death, spontaneous (nonprocedural) myocardial infarction, ischemic stroke, or ischemia-driven coronary revascularization) of 31% (HR 0.69; 95% CI 0.57–0.83; *p* < 0.001), which was due, again, to a significant reduction of MI (HR 0.7; 95% CI 0.53–0.93; *p* = 0.01) and also of ischemia-driven coronary revascularization (HR 0.75; 95% CI 0.60–0.94; *p* = 0.01) [142].

The analysis of the components of ACS in LoDoCo suggested that colchicine reduced the probability of acute coronary events unrelated to stent disease (i.e., in native segments), with lack of effect in the prevention of stent-related disease (i.e., acute stent thrombosis or stent restenosis) [141]. However, in a previous study that included diabetic patients undergoing PCI with bare metal stents, six-month angiographic restenosis rates were reduced by 62% in the group of patients randomized to colchicine as compared with patients in the control group (16% vs. 33%; OR 0.38, 95% CI 0.18–0.79; *p* = 0.007) [143]. These results may suggest that, along with atheroma plaque stabilization in native coronary arteries, the anti-inflammatory and anti-mitotic effects of colchicine may be equally effective in the prevention of neointimal hyperplasia, the central process in the pathophysiology of in-stent restenosis. However, the effects of colchicine preventing stent related disease in the era of new generation drug eluting stents may be unclear, as the rates of restenosis have decreased significantly [144].

As individual trials suffer from significant heterogeneity regarding the clinical setting (acute versus chronic CAD), treatment (colchicine dose and length of follow-up) and end point definitions, several systematic reviews and meta-analysis have been conducted to the summarize clinical effects of colchicine [145,146,147,148]. In one of these, by Fiolet and colleagues, inclusion criteria were restricted to RCT’s with a minimum follow-up of three months, thus including the five trials mentioned above. In their analysis, colchicine reduced the risk for the primary endpoint—a composite of MI, stroke, or cardiovascular death—by 25% (RR 0.75; 95% CI 0.61–0.92; *p* = 0.005) with a low between-trial heterogeneity (I^2^ = 23.9%). Colchicine led to a significant reduction of the individual endpoints MI by 22% (RR 0.78; 95% CI 0.64–0.94; *p* = 0.010), stroke by 46% (RR 0.54; 95% CI 0.34–0.86; *p* = 0.009), and coronary revascularization by 23% (RR 0.77; 95% CI 0.66–0.90; *p* < 0.001) [145]. In subgroup analysis the benefit observed with colchicine was consistent in both acute and chronic coronary syndrome and irrespective of gender [145]. It is noteworthy that the magnitude of the benefit obtained with colchicine in patients with CAD is comparable to that achieved by each of the mainstay therapies for the secondary prevention of CAD—such as antiplatelet agents and statins [149,150]—and has been achieved against a background of optimal treatment with these therapies.

The applicability of these findings may depend upon specific patient subsets. For example, the observed benefit of colchicine may be even higher in patients with diabetes mellitus. In a meta-analysis conducted by Kuzemczak and colleagues, in patients with CAD, the absolute risk reduction of the composite endpoint of MACE achieved by colchicine in patients with diabetes was greater than in patients without diabetes (absolute risk reduction of 3.94% vs. 2.32%, *p* < 0.001) [151]. On the other hand, as most trials have excluded patients with heart failure and chronic kidney disease, the effects of colchicine for the treatment of CAD in these important populations remain unknown.

Despite these favorable effects seen with colchicine, some trials have documented concerning results regarding non-CV mortality. In the COPS trial the rate of all-cause death was higher in the colchicine group compared with placebo (HR 8.20; 95% CI 1.03–65.61; *p* = 0.047) due to an increase in non-cardiovascular deaths, which were mostly due to sepsis [139]. Likewise, in the LoDoCo 2 trial there was an increase in non-cardiovascular deaths (HR 1.51; 95% CI 0.99–2.31), but without a parallel increase in severe infections, new cancer diagnosis or severe gastrointestinal adverse effects [142]. Meta-analyses have shown a non-significant lower incidence of cardiovascular mortality (RR 0.82; 95% CI 0.55–1.23; *p* = 0.339) counterbalanced by a non-significant higher incidence of non-cardiovascular deaths (RR 1.38; 95% CI 0.99–1.92; *p* = 0.060), with no difference in all-cause mortality (RR 1.08; 95% CI 0.71–1.62; *p* = 0.726) [145].

Taken together, RCTs show a consistent beneficial effect of colchicine by limiting new cardiovascular events and stroke. However, some barriers remain before a widespread use of colchicine in clinical practice can be adopted. Firstly, and as discussed above, its net effect on mortality is still under scrutiny, with a possible increase in non-cardiovascular deaths, which needs to be clarified in future trials. Secondly, in the era of precision medicine, a more individualized approach may be adopted to target specific populations where colchicine can produce a maximum benefit, such as in those patients with (i) persistent inflammation after the index event (i.e., persistently high hsCRP); (ii) particular markers of excess NLRP3 inflammasome activity (i.e., carriers of the rs10754555 gene variant); or (iii) high-risk of recurrence (such as diabetics). And finally, in the setting of secondary prevention of coronary artery disease, incorporating an additional drug to patients who are already under treatment with multiple medications with proven benefit brings forth the problem of poor adherence, as well as the risk of aggravating polypharmacy in an increasingly elderly and frail population. Table 2 summarizes the available phase 2 and phase 3 data on colchicine and atherosclerosis.

## 9. Conclusions

The inflammatory component of atherosclerosis pathogenesis offers new avenues through which novel therapies can be used and/or developed. Though around for decades, only recently has colchicine been in the eye of scientists and clinicians looking for new therapies for the management of coronary artery disease complications. The intracellular effects of colchicine directly impact key cellular players of inflammation, resulting in protective effects against atherosclerosis development (Figure 2). Translational research and phase 2 and 3 clinical trials have predominantly shown a beneficial effect of colchicine by modulating many underlying processes related to athero-inflammation and resulting in less clinical events. However, the net clinical effect upon mortality is still unclear and new trials must address this issue to be able to finally introduce this long-waited drug into the therapeutic toolkit to treat coronary artery disease.

## Figures and Tables

**Figure 1 pharmaceutics-14-01395-f001:**
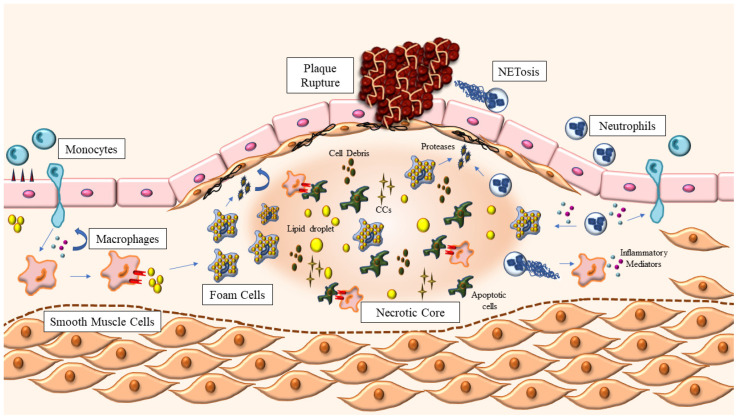
Atherosclerotic plaque development. Atherosclerosis starts with the accumulation of modified lipoproteins inside the vessel wall, which triggers the recruitment of leukocytes, monocytes and neutrophils from circulation. Once in the intima layer, monocytes differentiate into macrophages, which can now engulf the modified lipoproteins, becoming foam cells. Macrophages also continue to release inflammatory mediators—such as cytokines and chemokines—in response to the increased levels of cholesterol, further amplifying the response. Neutrophils also release pro-inflammatory mediators through granules and NETosis, contributing to an exacerbation of the inflammatory state within the vessel wall. Foam cells, apoptotic cells and cell debris, lipid droplets and extracellular cholesterol crystals (CCs) coalesce in the center of the growing plaque, forming the necrotic core, which is kept stable thanks to the fibrous cap: a structure made of smooth muscle cells and extracellular matrix proteins. The release of proteinases by macrophages and neutrophils weakens the fibrous cap, favoring plaque rupture and the exposure of the contents of the plaque to circulation, triggering blood coagulation and the clinical manifestations of atherosclerosis.

**Figure 2 pharmaceutics-14-01395-f002:**
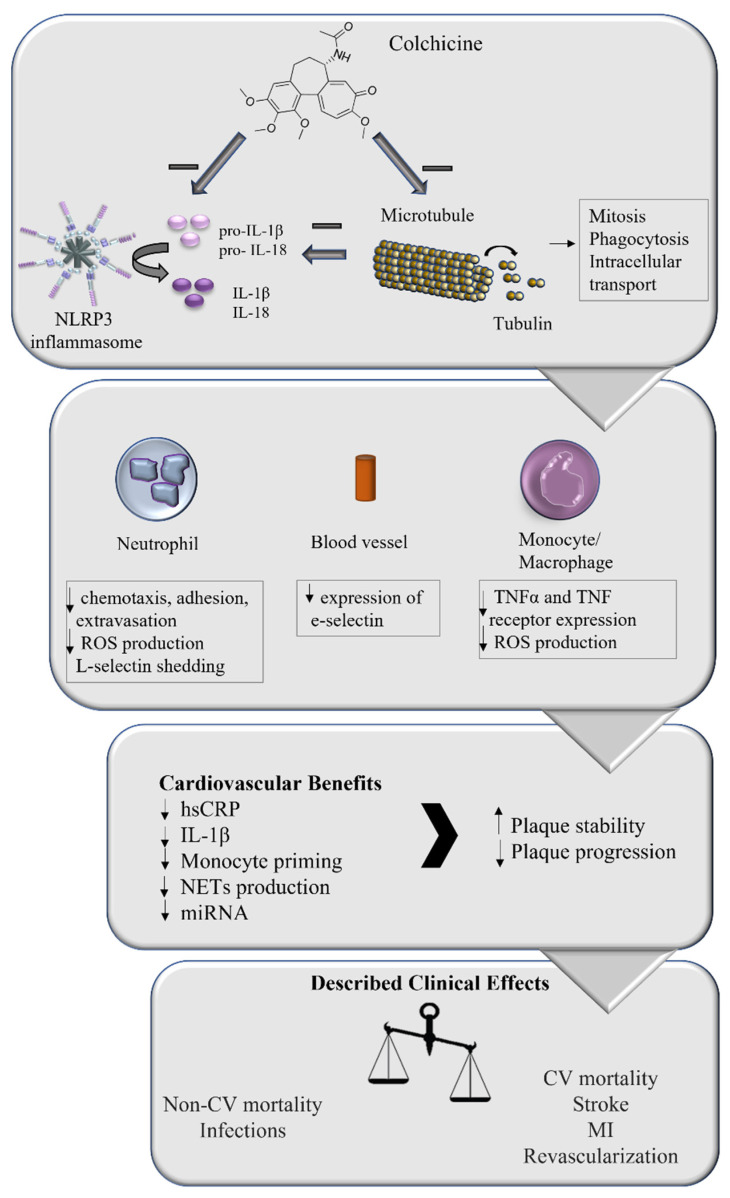
Role of colchicine in coronary artery disease treatment. Colchicine has been described to affect microtubule stability, impacting several intracellular processes including mitosis, phagocytosis, and intracellular transport. It has also been reported that colchicine affects NLRP3 inflammasome activation, impacting inflammatory cytokines production, both directly and through its action on microtubules. These intracellular effects directly impact the inflammatory response of neutrophils, monocyte/macrophages, and blood vessels, which translates into several cardiovascular benefits. The overall effect on plaque stability and progression impacts the clinical manifestations of atherosclerosis, reducing the incidence of major adverse cardiovascular effects, suggesting that the addition of colchicine to the management of coronary artery disease might be beneficial.

**Table 1 pharmaceutics-14-01395-t001:** Pre-clinical studies.

Study	Animal Model	Disease Induction	Colchicine Dosage *	Length of Intervention	Main Findings
Wojcicki et al., 1986 [107]	Rabbit	High-lipid diet	0.2 mg/kg i.p. twice a week	3 months	Reduction of circulating lipids, restoration of normal triglyceride levels and a protective effect on plaque development in the aorta
Lee et al., 1976 [108]	Yorkshire Swine	Balloon-induced denudation of aortic endothelium plus hypercholesterolemic diet	0.2 mg/kg/day	6 months	Slight worsening of atherosclerosis development in the aorta. No effect on serum cholesterol levels
Huang et al., 2014 [110]	Sprague–Dawley rats	High fat, high cholesterol diet for 6 weeks	0.5 mg/kg body weight/day i.p.	2 weeks	Reduction in circulating levels of C-reactive protein and lipoprotein associated phospholipase A2. Elevation of nitric oxide production. Effect was enhanced when administered along atorvastatin
Kaminiotis et al., 2017 [111]	New Zealand White rabbits	High cholesterol diet (1% *w*/*w*)	2 mg/kg body weight	7 weeks	No effect of colchicine on atherosclerosis or IL-18 levels. Slight effect on triglyceride levels
Spartalis et al., 2021 [112]	New Zealand White rabbits	High cholesterol diet (1% *w*/*w*)	2 mg/kg body weight plus 250 mg/kg body weight/day fenofibrate or 15 mg/kg body weight/day N-acetylcysteine (NAC)	7 weeks	Colchicine reduced aortic atherosclerosis especially when combined with NAC. Reduction in IL-6 and lower triglyceride levels were also reported
Mylonas et al., 2022 [113]	New Zealand White rabbits	High cholesterol diet (1% *w*/*w*)	2 mg/kg body weight plus 250 mg/kg body weight/day fenofibrate or 15 mg/kg body weight/day NAC	7 weeks	Reduction in de novo atherogenesis in the aorta and reduction of KLF4 expression in thoracic aortas
Akodad et al., 2017 [114]	C57BL/6 mice	Ligation of left coronary artery followed by reperfusion	400 μg/kg i.p.	25 min before reperfusion	Significant reduction of infarct size. Improvement of hemodynamic parameters. Decreased cardiac fibrosis
Mori et al., 2021 [115]	Wistar Rats	Ligation of left coronary artery followed by reperfusion	0.4 mg/kg/day i.p.	7 days	Reduction in post acute MI inflammation, ventricular remodeling, and dysfunction
Fujisue et al., 2017 [116]	C57BL/6J mice	Permanent ligation of left descending coronary artery	0.1 mg/kg/day	7 days port MI	Attenuation of pro-inflammatory cytokines and NLRP3 inflammasome components. Improved cardiac function, heart function and survival.
Cecconi et al., 2021 [117]	New Zealand White Rabbit	balloon endothelial denudation plus high cholesterol diet	0.2 mg/kg/day, 5 days/week, SQ	18 weeks	Reduction of the increase in aortic wall volume and inflammation

* Oral administration unless stated otherwise; i.p: intraperitoneal; NAC: N-acetylcysteine; NLRP3: nucleotide-binding oligomerization domain-like receptor, pyrin domain-containing 3; SQ: subcutaneous.

**Table 2 pharmaceutics-14-01395-t002:** Phase 2 and 3 clinical studies.

Trials	Setting	Key Inclusion Criteria	No. of Participants	Treatment	Main Results	Follow Up (Mean)
Giannopoulos et al. (2015) [127]	CABG	Patients undergoing CABG	59	Colchicine 0.5 mg BID vs. placebo	↓ 62% hsTnT and ↓ 52% CK-MB concentration	48 h after surgery
Deftereos et al. (2015) [128]	STEMI	STEMI ≤ 12 h from pain onset (treated with PCI)	151	Colchicine loading dose of 2 mg plus 0.5 mg BID vs. placebo	↓ 49% of CK-MB and ↓ 57% of hsTnT AUC concentration ↓ 25% MI volume (MRI)	9 days
COLIN (2017) [129]	STEMI	STEMI with one main coronary artery occluded	44	Colchicine 1 mg QD vs. placebo	No significant effect on: - hsCRP peak value	During hospitalization
Mewton et al. (2021) [130]	STEMI	STEMI referred for PCI	192	Colchicine 2 mg loading dose plus 0.5 mg BID vs. placebo	No significant effect on: - Infarct size at 5 days (MRI) - LV end-diastolic volume change at 3 months (MRI)	3 months
Vaidya et al. (2018) [131]	ACS	ACS (<1 month)	80	Colchicine 0.5 mg QD plus OMT vs. OMT alone	↓ Low attenuation plaque volume in CCTA (↓ 40.9% vs. ↓ 17%) hsCRP (↓ 37.3% vs. ↓ 14.6%)	12 months
Akrami et al. (2021) [132]	ACS	ACS (with medical therapy or PCI)	249	Colchicine 0.5 mg QD vs. placebo	↓ 71% MACE ↓ 84% ACS No significant effect on: - Decompensated HF - Death from any cause - Cardiovascular death	6 months
Fiolet et al. (2020) [133]	CCS	CCS and hsCRP ≥ 2 mg/L	138	Colchicine 0.5 mg QD	↓ 41% hsCRP levels ↓ 16% IL-6 levels	30 days
Colchicine-PCI (2020) [134]	PCI	Subjects referred for PCI (ACS or CCS)	400	Colchicine 1.8 mg pre-procedural	No significant effect on: - PCI-related myocardial injury - 30-day MACE (Death from any cause, MI, revascularization)	30 days
COPE-PCI (2021) [135]	PCI	Patients undergoing PCI (CCS or NSTEMI)	196	Colchicine 1.5 mg pre-procedural	↓ 41% Periprocedural myocardial injury	24 hrs
COLCOT (2019) [138]	ACS	MI (treated with PCI) within 30 days	4745	Colchicine 0.5 mg BID vs. placebo	↓ 23% MACE ↓ 84% Stroke ↓ 50% urgent hospitalization for angina leading to coronary revascularization No significant effect on: - Cardiovascular death - Resuscitated cardiac arrest - MI	19.5 months
COPS (2020) [139]	ACS	ACS treated with PCI or optimal medical therapy	795	Colchicine 0.5 mg BID for one month, then 0.5 mg QD for 11 months vs. placebo	↓ 84% Ischemia-driven urgent revascularization ↓ Death from any cause (8 vs. 1 patients) No significant effect on: - MACE - ACS - Stroke (ischemic, non-cardioembolic)	12 months
COPS (2021) [139]	ACS	ACS treated with PCI or optimal medical therapy	795	Same as above, no colchicine or placebo from months 13 to 24.	↓ 41% MACE ↓ 81% Ischemia-driven urgent revascularization No significant effect on: - Death from any cause - ACS - Stroke (ischemic, non-cardioembolic)	24 months
LoDoCo (2013) [141]	CCS	CCS, clinically stable for >6 months	532	Colchicine 0.5 mg QD vs. no colchicine	↓ 67% MACE ↓ 67% ACS No significant effect on: - Cardiac arrest - Stroke (ischemic, non-cardioembolic)	36 months
LoDoCo2 (2020) [142]	CCS	CCS, clinically stable for >6 months	5522	Colchicine 0.5 mg QD vs. placebo	↓ 31% MACE ↓ 30% MI (spontaneous, nonprocedural) ↓ 25% ischemia-driven coronary revascularization No significant effect on: - Cardiovascular death - Stroke (ischemic)	28.6 months
Deftereos S, et al. (2013) [143]	ACS/CCS	Diabetic patients undergoing PCI with BMS	196	Colchicine 0.5 mg BID vs. placebo	↓ 62% Angiographic in stent restenosis↓ 58% IVUS in stent restenosis	6 months

↓: indicates reduction of measured outcome. ACS: Acute coronary syndrome; AUC: area under the curve; BID: twice daily; BMS: bare-metal stent; CABG: coronary artery bypass grafting; CCS: chronic coronary syndrome; CCTA: coronary computed tomography angiography; CKMB: creatine kinase-MB; HF: heart failure; hsCRP: high-sensitive C reactive protein; hsTnT: high-sensitive Troponin T; IVUS: intravascular ultrasound; LV: left ventricle; MACE: Major adverse cardiovascular events, refers to the composite primary endpoint of each study, including all the individual outcomes listed in the box.; MI: myocardial infarction; MRI: magnetic resonance imaging; NSTEMI: non-ST-elevation MI; OMT: optimal medical therapy; PCI: percutaneous coronary intervention; QD: once daily; STEMI: ST-elevation MI; UA: unstable angina.

## Data Availability

Not applicable.

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
