# Peer review of "The Role of Colchicine in Atherosclerosis: From Bench to Bedside"

_pharmaceutics, 2022, doi:10.3390/pharmaceutics14071395_

Round 1
Reviewer 1 Report
In this review, the mechanisms and pharmacokinetics of colchicine were introduced. Particularly, the clinical data of colchicine for anti-inflammatory therapy are summarized. The structure of this manuscript is tight and balanced. In my opinion, this manuscript could be accepted for publication after minor revision.
1. Most of this review is in the introduction of colchicine in preclinical and clinical studies. More challenges in this field should be discussed. For example, what are the obstacles for the clinical applications of colchicine? What strategies could be used to enhance the application of colchicine in clinic.
2. Three-line table should be used in all Tables.
Reviewer 2 Report
The manuscript entitled "The role of Colchicine in Atherosclerosis. From bench to bed-side" by Leticia González et al is a nice effort to summarize the recent study related to colchicine and its anti-inflammatory application in cardiovascular disease. In this manuscript authors have detailed about the recent animal and human clinical studies related to atherosclerosis and coronary artery diseases. Authors have detailed every section with appropriate reference and examples. The introduction is short and crisp. Method is is well described and shows the seriousness of the study. Initially authors have shown the role of inflammation in atherosclerosis followed by detailed explanation of colchicine and NLRP3 inflammasome interaction. In spite of having some renal pathogenesis, colchicine shows high protein binding capacity. The next section talks about preclinical, clinical and translational studies of colchicine. Manuscript makes sense and reads well, especially the section VIII which could add a big effort in future studies related to colchicine. This section is very wisely written and explained and shows the in-depth knowledge of authors in the field. I request some minor modification if the author can address.
- Although all sections have been detailed I suggest adding a subsection of platelets and colchicine and its role in cardiovascular diseases to attract a wide subsets of readers. - As per current scenario it will be also useful to add some recent information if available about COVID, cardiovascular complications and colchicine. - Figure 1 The quality is very poor, please improve and some of the text is hard to read so please modify that too. - Line 423 "Mayor" correct this and please recheck the whole manuscript for any more typos.
Thank You
Reviewer 3 Report
Dear authors,
The role of Colchicine in Atherosclerosis. From bench to bedside.
is a very well written and narrated review. With the growing evidence of drug repurposing, colchicine fits into this category. Though this review is based on Atherosclerosis and Colchicine clinical data,
Structure of Colchicine, a paragraph of medicinal value of colchicine will add capture medicinal chemists, drug discovery teams.
The figure 1. font is very very small. Unable to read can you please increase the font.
- The introduction from line 70 - 100 you have talked about Atherosclerosis ---> inflammation and provided several biomarkers for the disease states. Can you pls. provide a figure to connect these biomarkers? It significantly helps the reader especially pharmacologists.
The table was very innovative with all the studies and compiled efficiently.
All the best.
Dr. Pashikanti.
